# Application of Fractal Dimension Analysis of Sublingual Blood Vessel Patterns in Correlation with Cardiovascular Diseases—A Pilot Study Title

**DOI:** 10.3390/jcm14051429

**Published:** 2025-02-20

**Authors:** Anastazja Janik, Kamil Jurczyszyn, Andrzej Wojtowicz, Fryderyk Zieliński, Mateusz Trafalski

**Affiliations:** 1Department of Dental Surgery, Medical University of Warsaw, St. Binieckiego 6, 02-097 Warszawa, Poland; anastazja.janik@wum.edu.pl (A.J.);; 2Department of Dental Surgery, Wroclaw Medical University, Krakowska 26, 50-425 Wroclaw, Poland; kamil.jurczyszyn@umw.edu.pl

**Keywords:** lingual blood vessels, aging, fractal dimension analyses, general health, cardiovascular diseases

## Abstract

**Background/Objectives**: The blood vessel network can be used as a potential marker of general health. In the oral cavity, it is possible to diagnose systemic diseases as the manifestations of many metabolic, cardiovascular, hematological, and autoimmune diseases may represent in lingual tissue. **Methods**: In the present study, a fractal dimension analysis (FD) of blood vessels of the tongue was applied as a marker of the physiology as well as pathology of cardiovascular diseases. This study was conducted for old men and women aged 68–95 years. Intraoral photography of the lower surface of the tongue was analyzed and correlated with cardiovascular diseases. **Results**: Differences in the value of the fractal dimension of blood vessels, between hypertension and the control group, were found, as well as in heart coronary disease, atherosclerosis, and heart valve defects. **Conclusions**: FD changes in the microvascular network of the ventral surface of the tongue can be regarded as potential markers of certain systemic diseases. Based on them, diseases with a cardiovascular basis can be identified, which may expedite the diagnostic process for patients or serve the long-term monitoring of these conditions.

## 1. Introduction

The oral cavity, being the initial segment of the digestive tract, plays a crucial role not only in the process of digestion but also in the diagnosis of numerous systemic diseases. The manifestations of these diseases in the oral cavity can provide valuable diagnostic clues, which is particularly important for dentists and internal medicine specialists. Many metabolic, autoimmune, infectious, hematological, and cardiovascular diseases exhibit their initial symptoms in this part of the body.

One of the groundbreaking achievements in medicine is the utilization of non-invasive methods for imaging blood vessels. Images of vessels obtained from examinations such as ophthalmoscopy, angiography, and OCTA (optical coherence tomographic angiography) are most commonly subjected to analysis and widely used in ophthalmology, cardiology, radiology, and oncology [1,2,3,4,5]. However, the network of blood vessels on the ventral surface of the tongue has not been extensively described thus far. Therefore, these vessels may become another promising marker for the diagnosis and evaluation of systemic diseases. In the present study, fractal dimension (FD) analysis was utilized for imaging lingual vessels. An attempt was made to demonstrate the relationship between selected cardiovascular diseases (CVDs) and the FD of large and small blood vessels located on the ventral surface of the tongue. Additionally, the association between certain peripheral blood parameters and the morphology of these vessels was investigated, utilizing fractal dimension analysis as well.

A fractal is a complex geometric shape that can be split into parts, each of which is a reduced-scale copy of the whole. This property is known as self-similarity. Fractals are infinitely complex, meaning they exhibit detail at every level of magnification. In mathematics, fractals are described by fractal dimensions, which provide a statistical index of complexity comparing how detail in a pattern changes with the scale at which it is measured.

Fractal analysis has been employed to assess the morphology of sublingual blood vessels, particularly in relation to cardiovascular disease (CVD). This methodology aims to characterize the complexity and irregularity of vessel shapes, offering insights into vascular health and disease progression. By examining the fractal dimension and patterns of sublingual microcirculation, researchers can potentially identify early markers of CVD and monitor disease development. Understanding the fractal properties of blood vessels contributes to advancements in diagnostic approaches and therapeutic interventions for cardiovascular conditions.

In classical, Euclidian geometry, we are used to thinking that dimension is an integer value. For instance: the 0 is a dimension of point; a section has one dimension—length, whereas length and width describe plane figures, and solids have three dimensions, length, width, and height. Fractals go beyond these rules. Their dimensions are rational numbers and may take values between 0 and 3. Self-similarity is another feature of fractals. It seems that in any scale of observation, fractals look similar. Benoît Mandelbrot is the father of fractals. It is easy to describe the shapes of simple figures. Nature, especially living forms, is full of shapes that may be treated as fractals up to a certain scale, such as neuron networks and nets of blood vessels. It is hard to describe these shapes using simple Euclidian geometry. It is possible to measure area or perimeter, but a more complex shape needs to be detailed, and it is more difficult to do so. Calculation of the fractal dimension (FD) enables obtaining a fractional number that describes the examined shape. It is a relation between complexity and the examined shape. The more complex it is, the lower the value of the FD. An example of this relationship is shown in Figure 1.

FD imaging of the tongue’s vessels may become a useful tool for assessing and monitoring the most common cardiovascular diseases. To our knowledge, the study we conducted is the first to describe this association. To fully realize the potential of this method, it is necessary to refine the standards of this procedure by extending it to long-term studies on a larger patient group.

The aim of this study was the application of fractal dimension analysis to find any connections between a picture of sublingual vessels and general diseases, especially cardiovascular diseases.

We put forward the following null hypotheses:

There are no differences in the value of the fractal dimension of large sublingual vessels between the control and examined group;There are no differences in the value of the fractal dimension of capillary vessels between the control and examined group;There are no differences in the value of the fractal dimension of large sublingual vessels between the examined diseases and control group;There are no differences in the value of the fractal dimension of capillary vessels between the examined diseases and control group.

## 2. Materials and Methods

### 2.1. Patients

The examined group comprised 35 patients, with a mean age of 84 years (SD = 7). Residents of the Nursing Home of the Sisters of Charity of St. Anthony in Czestochowa in 2021 and 2022 qualified for the study, which was conducted in the 68–95 years age group, consisting of 7 men and 28 women. The research team comprised two people, an examiner and dental assistant, for the purpose of registering data and preparing the workplace and tools. The examination of patients included a medical interview (concerning systemic diseases and medications) and an intraoral examination, which was taken using dental instruments (flat mirror) and the Identafi Dental EZ device, Malvern, PA, USA) in a complete darkroom. For imaging purposes, the tongue surface was dried using sterile gauze. Patients were instructed to open their mouths widely and extend the tip of their tongue toward the hard palate. The light source was directed at the central area of the target from a distance of approximately 3.5 cm. General diseases included hypertension, coronary artery disease, atherosclerosis, atrial fibrillation, and heart valve defects. There were no exclusion criteria for the study group from the nursing home.

A total of 31 individuals participated in the control group study, consisting of 16 women and 15 men, all of Caucasian ethnicity, aged between 18 and 46 years. These were patients who presented for general dental services, as well as employees of the University Dental Center in Warsaw who voluntarily joined the control group study. The inclusion criteria for the control group were an absence of systemic diseases, no regular medication intake, and no history of tobacco or nicotine product use.

The project was approved by the Bioethics Committee of the Regional Medical Chamber in Częstochowa, approval number K.B.Cz.-0005/2022 (20 April 2022).

### 2.2. Taking Photos and Region of Interest Selection

All photos were taken using an iPhone 12Pro (Apple Inc., Cupertino, CA, USA). The base resolution of images was 3024 × 4032 pixels. To achieve repeatability, we used the same distance between the lens and vessels. The optical axis of the lens and oral mucosa was approximately 90 degrees to reduce the risk of reflection. We took photos in the full spectrum of light (white light) and green light using the Identafi Dental EZ lamp, Malvern, PA, USA. White light (highly concentrated whiteness) provides an optimal image of the oral mucosa in a conventional examination. Green light illuminates the blood vessels, allowing the clinician to find areas of dilated vascularity.

In green light, the capillary tubes of the mucous membrane were visible. In white light, large sublingual vessels were better seen. The size of the regions of interest (ROIs) was set at 250 × 250 pixels. We excluded sites where reflections occurred. In the case of white light, image ROIs were selected at the border of sublingual vessels and the mucous membrane. In some patients in the examined groups, additional vessels were observed. These ROIs were collected from white light images and analyzed separately. In green light photography, ROIs were selected from the mucous which covers the sublingual vessels with images of capillary tubes. All ROIs were converted to 8-bit monochromatic images, which were the basis of the fractal dimension analysis. The procedure of taking ROIs is seen in Figure 2. Generally speaking, the white light FD describes the shape of large sublingual vessels, whereas the green light FD describes the capillary tubes of the mucous membrane in the sublingual region. The FD of white light additionally describes the shapes of vessels that occurred at the periphery of the sublingual region (yellow squares in Figure 1A). The mean number of ROIs per patient was 8.6. All graphical operations were performed using Adobe Photoshop 7 (Adobe Inc., San Jose, CA, USA).

### 2.3. Fractal Dimension Analysis

ImageJ version 1.53e (Image Processing and Analysis in Java—Wayne Rasband and contributors, National Institutes of Health, Bethesda, MD, USA, public domain li-cense, https://imagej.net/ij/, accessed on 1 January 2024) and the FracLac plugin version 2.5 (Charles Sturt University, Bathurst, Australia, public domain license) were used to perform all fractal analyses. We applied the intensity difference fractal dimension counting method with block analysis. This algorithm allows us to analyze 8-bit monochromatic images. It was fully described in our previous study [6].

### 2.4. Statistical Analysis

Statistica version 13.3 (StatSoft, Krakow, Poland) was used to perform all statistical tests. A statistical significance level of 0.05 was assumed. The Shapiro–Wilk test was applied to confirm the normality of the distribution. Due to a normal distribution, we performed a *t*-Student parametric test (for independent samples) to check the statistical differences between the control and examined groups with respect to fractal dimension. The power of each statistical test was calculated on the basis of the number of samples and the mean and standard deviation of all examined groups. It was calculated for alpha = 0.05 and for a two-sided null hypothesis.

## 3. Results

Figure 3 shows a comparative presentation of the images from the control and disease groups with examples of the fractal dimension values for selected regions of interest.

The distribution of diseases is shown in Table 1. Hypertension was the most common observed disease (19 patients), followed by coronary disease, which was seen in 10 patients, atherosclerosis in 9 cases, atrial fibrillation in 5 cases, and heart valve defects in 4 patients.

The results of the FD analysis of large and capillary vessels are presented in Table 2. A statistically significant difference was observed between the examined and control groups in regard to large sublingual and capillary vessels. Important to underline is the fact that the fractal dimension value of large sublingual vessels was higher than in the control group in contrast to the FD of capillary vessels. The FD of capillary vessels was lower than in the control group. In this case, the difference of the FD value of capillary vessels versus the control group is on the edge of statistical significance (*p* = 0.037).

The results of the *t*-Student test for the fractal dimension of the large sublingual vessels of the examined diseases versus the control group are shown in Table 3. It is important that the statistical differences in the fractal dimension value of specific diseases versus the control group are seen in all of the diseases: hypertension, heart coronary disease, atherosclerosis, atrial fibrillation, and heart valve defects versus the control group. The highest value of FD is noted in the case of atrial fibrillation (1.4414); in contrast, the lowest FD value is in the case of hypertension (1.3830). The standard deviation is similar, in the range between 0.0501 and 0.0819. In all of the cases, the FD value was higher than in the control group.

The results of the *t*-Student test for the fractal dimension of capillary vessels of the examined specific diseases versus the control group are shown in Table 4. In this aspect, a statistical difference was observed only in the case of hypertension. The rest of the examined diseases did not reveal significant differences versus the control group.

## 4. Discussion

The oral cavity is easy to visualize, which provides quick and non-invasive imaging of the tongue. It has long been known that the tongue can be a site of manifestation for many systemic diseases such as celiac disease, GERD, Crohn’s disease, eating disorders, HIV, and asthma [7]. Pathologies of the tongue most commonly manifest as geographic tongue, swelling (edema), discolorations, changes in color, coated tongue, ulcers, or blisters [8,9]. However, until now, few attempts have been made to investigate the relationship between systemic diseases and the vascular image of the ventral side of the tongue. It is known that with age, the incidence of sublingual varices (SVs) increases, which are located on the ventral and lateral sides of the tongue [10]. The underlying cause of this disorder may be the weakening of the venous walls due to the degeneration of elastic fibers associated with aging [11]. Kleinman’s study indicates that the presence of sublingual varices is strongly correlated with the aging process, and their appearance before the fifth decade of life may indicate premature aging of the organism [12]. Our study results confirm this fact, as all patients (average age 84 years, SD = 7) showed the presence of enlarged large vessels in the tongue.

Numerous studies indicate a correlation between the presence of sublingual varices (SV) and hypertension [10,13,14,15,16,17].

In the case of hypertension, we observed statistical differences in the fractal dimension (FD) for both large and capillary vessels when compared to the control group. Specifically, the mean FD for large vessels in the control group is lower than in the examined group, whereas for capillary vessels, the mean FD in the control group is higher than in the examined group. The classic interpretation of the fractal dimension value indicates that the higher the FD (closer to 2), the less complex the shape. Our results show that in the case of hypertension, the structure of large sublingual vessels is less complex than in the control group. Conversely, for capillary vessels, the structure is more complex in the diseased group compared to the control group.

Hedström et al. report in their study that the occurrence of SVs is significantly associated with age, smoking, and cardiovascular diseases, mainly hypertension [13]. Furthermore, they state that the relationship between arterial hypertension and the enlargement of sublingual vessels is more pronounced in smokers with hypertension than in non-smokers with this condition. In this study, image analysis was based solely on the evaluation of photographs by two independent researchers, which may introduce bias and affect the results. In our studies, subjective assessment was eliminated in favor of objective fractal dimension analysis. Akkaya et al. report that the risk almost doubles when patients with arterial hypertension use removable dentures [10]. However, in this study, only one researcher assessed the vessels using the Hedström and Bergh two-point scale (absence/mild SVs—grade 0 and present/severe SVs—grade 1), which may also affect the objectivity of the results. Accardo et al. also demonstrated an association between both controlled and resistant hypertension and SVs [18]. They did not reveal such an association with newly diagnosed hypertension, but this relationship was visible in the group of patients with dyslipidemia and newly diagnosed hypertension. Nonetheless, in the entire project, the sublingual vessels were assessed only visually during the clinical examination using the aforementioned two-point scale, which may significantly distort the study results. Shivakumar et al. used the same scale in their studies and also demonstrated an association between hypertension and the presence of tongue varices [15]. Furthermore, they showed that the frequency of SVs increases with the degree of hypertension. In this case, the vessels were also assessed only visually during the examination, which may involve similar risks as in the previously mentioned studies.

However, Kleinman states that SVs (sublingual varices) are not interdependent with pulmonary diseases and cardiovascular diseases [12]. However, this remains in opposition to our observations, as we have demonstrated the association between the shape of large sublingual vessels and cardiovascular diseases. In Kleinman’s studies, the assessment was conducted by four independent researchers, evaluating the tongue only during the study, which may negatively impact the results. Therefore, by applying FD for the image analysis, such risk can be excluded, ensuring research objectivity.

Another clinical approach was characterized by the study of Bergh et al. [16]. They conducted an eight-year observation of patients with arterial hypertension regarding the presence and development of SVs. The state of sublingual vessels was assessed during annual examinations using the aforementioned two-stage scale. This study showed that in 76.5% of participants, the condition of sublingual varices remained unchanged over 8 years, and the development from stage 0 to stage 1 was mainly associated with advanced age. In individuals where stage 1 developed during the observation period, the average age was higher, and cardiovascular diseases were more common.

Khalilzada et al. attempted to demonstrate the relationship between sublingual microcirculation and small vessel disease (SVD) of the brain [19]. Their research involved analyzing the dynamic image of small sublingual vessels captured by a handheld dark-field microscopy camera with a side-stream light at 530 nm and a system of objective lenses with a fivefold magnification (MicroVision Medical, Amsterdam, The Netherlands). They assessed the microvascular flow index and the functional density of sublingual capillaries. Researchers observed a lower functional density of capillaries along with an abnormal flow in patients with SVD. They indicate a generalized microcirculation disorder in SVD patients, observable in the sublingual area. The exact cause of this condition is not fully understood, but authors suggest it may result from the temporary occlusion or obliteration of arterioles or capillaries. Thus, they demonstrate that the cerebral form of SVD could be part of a generalized vascular disorder, evident also on the ventral side of the tongue.

Regarding the other cardiovascular diseases (excluding arterial hypertension) included in our project (coronary artery disease, arteriosclerosis, artificial heart valves, atrial fibrillation, heart valve defects), we also observed statistical differences in the FD range of large vessels compared to the control group. However, we did not demonstrate (as was the case with arterial hypertension) a difference in the FD range of capillary vessels. In this instance, the average FD of large vessels in the examined group is significantly higher (range 1.4078–1.4414) than in arterial hypertension (1.3830) and the control group (1.3282). Our research findings indicate that concerning various cardiovascular diseases, the image of large sublingual vessels is less complex than in the control group. However, in the FD range of capillary vessels, we did not demonstrate a statistical difference between the aforementioned diseases (except for arterial hypertension) and the control group.

In the literature, there is limited evidence describing the association between the shape of sublingual vessels and these diseases. Most commonly discussed is arterial hypertension, generally classified as a cardiovascular disease. Few studies conducted in this direction differentiate other conditions within this group, such as atrial fibrillation, heart valve insufficiency, or coronary artery disease. Research by Al-Shayyab et al. indicates that the risk of SV occurrence is nearly four-times higher in individuals suffering from cardiovascular diseases [20]. In the group of patients with cardiovascular disease (CVD), hypertension, angina pectoris, myocardial infarction, stroke, and heart valve dysfunctions were distinguished. Their results did not describe each condition separately but rather all were included in one group of cardiovascular diseases. Such an approach, along with observations based solely on language examination and classification of vessel pathology on a two-step scale, may affect the objectivity of the results.

Bergh et al. analyzed the association between ischemic heart disease (IHD) and arterial hypertension with the frequency of SV occurrence [21]. Ischemic heart disease was classified into myocardial infarction and angina pectoris, further categorized as newly diagnosed (revealed during ongoing observations) or pre-existing (diagnosed before the study commencement). Sublingual veins were evaluated on a two-step scale (no/few SVs or moderate/severe SVs) by analyzing images of the right and left sides of the lingual frenulum. Assessment was conducted by two researchers utilizing double-blind methodology. They demonstrated that newly diagnosed IHD occurs more frequently in the SV group than in pre-existing IHD. The authors suggest a temporal relationship wherein SVs develop shortly before IHD. Due to the limited sample size and the methodology of language assessment, such a hypothesis should be cautiously interpreted.

Another area of the human body where blood vessels are easily visualized is the retina. Numerous studies have shown a correlation between retinal vessel patterns and cardiovascular diseases. Fractal dimension (FD) analysis is often used to assess the architecture of retinal vessels. Cheung et al. demonstrated a lower fractal dimension of retinal vessels in older patients with hypertension and diabetes compared to healthy individuals [22]. Similar observations were made by Liew et al., who compared older patients with hypertension to healthy patients [23]. In other studies, Liew et al. showed a correlation between a lower FD of retinal vessels and higher mortality due to stroke [24]. They suggest that the reduced fractal dimension of the retinal arterioles is associated with microbleeds in the brain. Our studies also showed a lower value of the fractal dimension of capillaries, but only in the case of hypertension. In the case of large sublingual vessels, the fractal dimension was higher in all the diseases studied than in the control group. Therefore, further studies are needed, especially studies comparing the fractal dimension of retinal and sublingual vessels in systemic diseases.

Despite numerous reports on the association of cardiovascular disease (CVD) with SVs, the mechanism of their occurrence has not been unequivocally established. Nevertheless, a simple and common examination of the lingual frenulum’s ventral and lateral sides may contribute to the detection of cardiovascular diseases, especially arterial hypertension. This facilitates the prompt implementation of specialized treatment, significantly reducing the risk of serious complications. Therefore, dentists may encounter symptoms of these conditions first and play a significant role in the diagnostic process.

## 5. Conclusions

There are statistical differences between the study group and the control group in relation to the fractal dimension of large sublingual vessels. Null hypothesis rejected.There are statistical differences between the examined group and the control group in relation to the fractal dimension of capillary vessels, but the *p* value is on the edge of statistical significance. Null hypothesis rejected.There are differences in the value of the fractal dimension of large sublingual vessels between the examined diseases and the control group in all of the diseases: hypertension, heart coronary disease, atherosclerosis, atrial fibrillation, and heart valve defects. Null hypothesis rejected.There are differences in the value of the fractal dimension of capillary vessels between hypertension and the control group. Null hypothesis rejected.Our results show that the fractal dimension analysis of sublingual vessels can be used in screening programs.

FD changes in the microvascular network of the ventral surface of the tongue can be regarded as potential markers of certain systemic diseases. Based on them, diseases with a cardiovascular basis can be identified, which may expedite the diagnostic process for patients or serve the long-term monitoring of these conditions. Therefore, a simple and common examination of the ventral and lateral aspects of the tongue during routine dental examinations may contribute to the detection of these diseases. This facilitates the prompt implementation of specialized treatment, significantly reducing the risk of serious complications.

## 6. Study Limitations

In the examined group, some patients suffered from more than one disease, which potentially could have influenced the value of the fractal dimension.One limitation is that the study only looks back in time; it cannot prove cause and effect, only associations.Small group sizes but data increased by examination of more regions of interests (ROIs) than patients.Discrepancy in age between the control and examined groups may affect our results, but creating a control group without any cardiovascular diseases in similar age to the examined group is almost impossible.It is a retrospective study design, so causality could not be established, only associations.

## Figures and Tables

**Figure 1 jcm-14-01429-f001:**
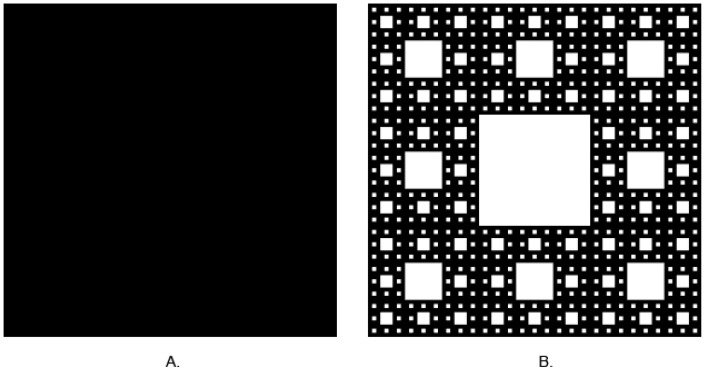
(**A**) Square—FD = 2, (**B**) Sierpinski carpet—FD ≈ 1.8928 (Generated by https://codinglab.huostravelblog.com/math/fractal-generator/ accessed on 1 January 2024).

**Figure 2 jcm-14-01429-f002:**
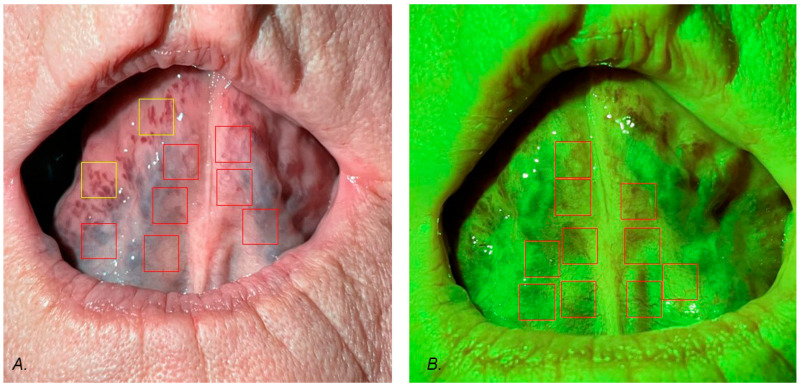
Region of interest (ROI) selection: (**A**) white light photography, (**B**) green light photography, red squares—ROIs, yellow squares—additional ROIs.

**Figure 3 jcm-14-01429-f003:**
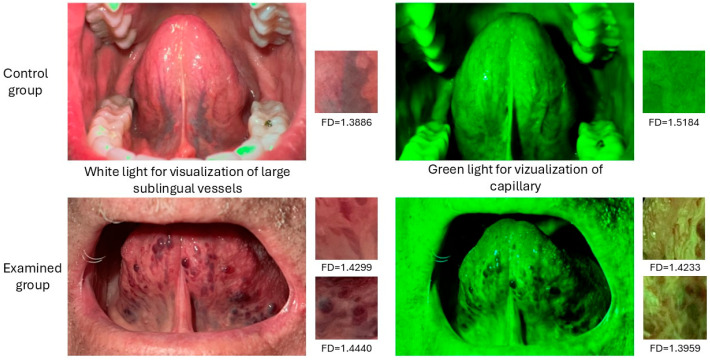
Comparative presentation of photos from the control and disease groups with examples of the fractal dimension values for selected regions of interest.

**Table 1 jcm-14-01429-t001:** Amount of occurrence of specific diseases.

Disease	Hypertension	Heart Coronary Disease	Atherosclerosis	Atrial Fibrillation	Heart Valve Defects
Amount of	19	10	9	5	4

**Table 2 jcm-14-01429-t002:** Results of the *t*-Student test for the fractal dimension of ROIs in white and green light (FD—value of fractal dimension, *t*—value of *t*-Student test, df—degrees of freedom, *p*—*p* value, SD—standard deviation, N—number of examined ROIs).

	Power of Test	Mean (1) Examined Group	Mean (2) Control Group	*t*	df	*p*	SD (1)	SD (2)
FD of large sublingual vessels (N = 42)	0.987	1.4019	1.3282	6.06	198	0.000000	0.07340	0.08619
FD of capillary vessels (N = 193)	0.764	1.4184	1.4369	−2.09	288	0.037463	0.06788	0.07640

**Table 3 jcm-14-01429-t003:** Results of the *t*-Student test for the fractal dimension of large sublingual vessels of the examined specific diseases versus the control group (FD—value of fractal dimension, *t*—value of *t*-Student test, df—degrees of freedom, *p*—*p* value, SD—standard deviation).

	Power of Test		Mean (1) Examined Group	Mean (2) Control Group N = 119	*t*	df	*p*	SD 1	SD 2
FD of large sublingual vessels	0.999	Hypertension (N = 91)	1.3830	1.3282	4.1522	145	0.0001	0.0721	0.0862
	1.000	Heart coronary disease (N = 73)	1.4078		5.3464	127	0.0000	0.0819	
	0.999	Atherosclerosis (N = 60)	1.4156		5.7255	114	0.0000	0.0782	
	0.999	Atrial fibrillation (N = 23)	1.4414		5.8904	77	0.0000	0.0501	
	0.999	Heart valve defects (N = 20)	1.4357		5.1618	74	0.0000	0.0581	

**Table 4 jcm-14-01429-t004:** Results of the *t*-Student test for the fractal dimension of capillary vessels of examined specific diseases versus the control group (FD—value of fractal dimension, *t*—value of *t*-Student test, df—degrees of freedom, *p*—*p* value, SD—standard deviation, N—number of examined ROIs).

	Power of Test		Mean (1) Examined Group	Mean (2) Control Group N = 119	*t*	df	*p*	SD 1	SD 2
FD of capillary vessels	0.951	Hypertension (N = 134)	1.4031	1.4369	−3.6885	229	0.0003	0.0623	0.0764
	0.320	Heart coronary disease (N = 76)	1.4182		−1.5334	171	0.1270	0.0834	
	0.050	Atherosclerosis (N = 61)	1.4374		0.0486	156	0.9613	0.0687	
	0.247	Atrial fibrillation (N = 25)	1.4572		1.2407	120	0.2171	0.0569	
	0.054	Heart valve defects (N = 25)	1.4333		−0.2038	120	0.8389	0.0790	

## Data Availability

Data are available from the authors at anastazja.janik@wum.edu.pl, andrzej.wojtowicz@wum.edu.pl.

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
