# Peer review of "Application of Fractal Dimension Analysis of Sublingual Blood Vessel Patterns in Correlation with Cardiovascular Diseases—A Pilot Study Title"

_jcm, 2025, doi:10.3390/jcm14051429_

Round 1
Reviewer 1 Report
Comments and Suggestions for Authors
The method of the manuscript is well described, and the manuscript is both well-structured and thoroughly discussed. The authors have also acknowledged some limitations of the study.
Major Issue:
A significant concern is the composition of the control group. In addition to the small sample size of both the patient and control groups, the control group consists of young and healthy individuals, which contrasts significantly with the patient group. This introduces a critical confounding factor, as the observed significant differences in the fractal dimension (FD) of the tongue’s vessels could potentially result from age-related vascular changes rather than being solely attributable to cardiovascular diseases.
Minor issues:
1,Line 194: The statement, “The highest value of FD is noted in the case of heart valve defects (1.4357),” is inconsistent with the data presented in Table 3. The table indicates that atrial fibrillation has the highest FD value (1.4414). This discrepancy needs to be corrected for accuracy.
2,Tables 2, 3, and 4: The legends state that significant p-values are indicated by red color (<0.05) and underlining (p < 0.05). However, these visual indicators are absent from the tables. This inconsistency should be addressed to ensure clarity and alignment with the stated legend.
Author Response
Major issue:
A significant concern is the composition of the control group. In addition to the small sample size of both the patient and control groups, the control group consists of young and healthy individuals, which contrasts significantly with the patient group. This introduces a critical confounding factor, as the observed significant differences in the fractal dimension (FD) of the tongue’s vessels could potentially result from age-related vascular changes rather than being solely attributable to cardiovascular diseases.
Response:
We are aware of the age discrepancy between the control group and the study group. This discrepancy results from the difficulty in finding healthy patients of a similar age to the study group. This is the main reason for the significant age difference between the two groups. We agree that this discrepancy may have influenced some of the results, and this information has been added to the "Study Limitations" section. The small number of patients who participated in the study was compensated by the number of regions of interest examined. We have added the number of regions of interest to each table. To confirm the sufficiency of the number of areas of interest and to obtain reliable results, the power of each statistical test was calculated. The power value for the test has been added to each table. The relevant information has also been added to the "Materials and Methods" section.
Minor issues:
- Line 194: The statement, “The highest value of FD is noted in the case of heart valve defects (1.4357),” is inconsistent with the data presented in Table 3. The table indicates that atrial fibrillation has the highest FD value (1.4414). This discrepancy needs to be corrected for accuracy.
Response:
It was corrected and yellow highlighted in the manuscript.
- Tables 2, 3, and 4: The legends state that significant p-values are indicated by red color (<0.05) and underlining (p < 0.05). However, these visual indicators are absent from the tables. This inconsistency should be addressed to ensure clarity and alignment with the stated legend.
Response:
Table legends were corrected, and p-values lower than 0.05 were underlined.
Best regards,
Authors
Reviewer 2 Report
Comments and Suggestions for Authors
The submitted paper suggests that fractal dimension (FD) analysis of the tongue's blood vessel network can be a useful marker for cardiovascular diseases in older adults. The study highlights significant differences in FD values between individuals with conditions like hypertension, coronary heart disease, atherosclerosis, and heart valve defects compared to a control group. The findings imply that FD changes in the tongue's microvascular network could aid in early diagnosis and long-term monitoring of systemic diseases with a cardiovascular basis. The paper is overall well written, topic is of interest, and I do not have any additional comment.
Author Response
The submitted paper suggests that fractal dimension (FD) analysis of the tongue's blood vessel network can be a useful marker for cardiovascular diseases in older adults. The study highlights significant differences in FD values between individuals with conditions like hypertension, coronary heart disease, atherosclerosis, and heart valve defects compared to a control group. The findings imply that FD changes in the tongue's microvascular network could aid in early diagnosis and long-term monitoring of systemic diseases with a cardiovascular basis. The paper is overall well written, topic is of interest, and I do not have any additional comment.
Response:
Thank you for the positive review.
Best regards,
Authors
Reviewer 3 Report
Comments and Suggestions for Authors
Thank you for allowing me to review your paper. I understand that the study investigates whether fractal dimension (FD) analysis of sublingual blood vessels can serve as a marker for cardiovascular diseases (CVDs). It has several strengths: it is the first study to apply FD analysis to sublingual vasculature for CVD screening, it eliminated subjective visual assessments used in prior studies, and it has the potential for integration into routine dental exams.
However, here are a few suggestions and minor changes that can be made to improve the quality of the papers at the author's discretion:
1. Control group (18–46 years) vs. CVD group (68–95 years). Age-related vascular changes (e.g., sublingual varices) were not adjusted for confounding results.
2. Small sample sizes, particularly for specific CVDs (e.g., 4 patients with heart valve defects), reduce statistical power.
3. Multiple comparisons without correction increases the risk of Type I errors.
4. Conclusion #2 incorrectly states a sustained null hypothesis despite p = 0.037 (statistically significant). Consider rephrasing the structure of the sentence.
5. Given that retrospective design cannot establish causality, only associations.
6. In Ref. 3: Author "Utsunomiya" lacks initials.
Author Response
Thank you for the valuable suggestions.
- Control group (18–46 years) vs. CVD group (68–95 years). Age-related vascular changes (e.g., sublingual varices) were not adjusted for confounding results.
We are aware of the age discrepancy between the control group and the study group. This discrepancy arises from the difficulty in finding healthy patients of a similar age to the study group. This is the main reason for the significant age difference between the two groups. We agree that this discrepancy may have influenced some of the results, and this information has been added to the "Study Limitations" section.
- Small sample sizes, particularly for specific CVDs (e.g., 4 patients with heart valve defects), reduce statistical power.
The small number of patients who participated in the study was compensated by the number of regions of interest examined. We have added the number of regions of interest to each table. To confirm the sufficiency of the number of regions of interest and to obtain reliable results, the power of each statistical test was calculated. The power value for the test has been added to each table. The relevant information has also been added to the "Materials and Methods" section.
- Multiple comparisons without correction increases the risk of Type I errors.
The power value for the test has been added to each table. The relevant information has also been added to the "Materials and Methods" section.
- Conclusion #2 incorrectly states a sustained null hypothesis despite p = 0.037 (statistically significant). Consider rephrasing the structure of the sentence.
It was corrected.
- Given that retrospective design cannot establish causality, only associations.
We added it to the "Study Limitations."
- In Ref. 3: Author "Utsunomiya" lacks initials.
Corrected.
Best regards,
Authors
Round 2
Reviewer 1 Report
Comments and Suggestions for Authors
Thank the authors' response and revisions. I have no further comments on the manuscript.